# Shuttling single metal atom into and out of a metal nanoparticle

Shuxin Wang[1,2], Hadi Abroshan [1], Chong Liu [3], Tian-Yi Luo[3], Manzhou Zhu [2], Hyung J. Kim [1,4], Nathaniel L. Rosi[3] & Rongchao Jin[1]

It has long been a challenge to dope metal nanoparticles with a specific number of hetero-metal atoms at specific positions. This becomes even more challenging if the heterometal belongs to the same group as the host metal because of the high tendency of forming a distribution of alloy nanoparticles with different numbers of dopants due to the similarities of metals in outmost electron configuration. Herein we report a new strategy for shuttling a single Ag or Cu atom into a centrally hollow, rod-shaped $Au_{24}$ nanoparticle, forming $AgAu_{24}$ and $CuAu_{24}$ nanoparticles in a highly controllable manner. Through a combined approach of experiment and theory, we explain the shuttling pathways of single dopants into and out of the nanoparticles. This study shows that the single dopant is shuttled into the hollow $Au_{24}$ nanoparticle either through the apex or side entry, while shuttling a metal atom out of the $Au_{25}$ to form the $Au_{24}$ nanoparticle occurs mainly through the side entry.

[1] Department of Chemistry, Carnegie Mellon University, Pittsburgh, PA 15213, USA. [2] Department of Chemistry and Center for Atomic Engineering of Advanced Materials, Anhui University, Hefei 230601, China. [3] Department of Chemistry, University of Pittsburgh, Pittsburgh, PA 15213, USA. [4] School of Computational Sciences, Korea Institute for Advanced Study, Seoul 02455, Korea. Shuxin Wang and Hadi Abroshan contributed equally to this work. Correspondence and requests for materials should be addressed to R.J. (email: rongchao@andrew.cmu.edu)

Nanoparticles play a central role in the rapidly growing nanoscience and nanotechnology fields, with a wide range of applications being developed including nanocatalysis, sensing, optical, and biology[1–5]. Atomic level understanding of nanoparticle structure is of great importance in order to establish definitive structure—property relationships[6], thereby facilitating systematic tailoring of material properties and developing of various applications of nanoparticles[1–3]. In this regard, atomically precise gold nanoparticles have attracted great interest in recent years for both fundamental research and technological applications[1]. Recent success in the synthesis of atomically well-defined nanoparticles[6–10] has offered exciting opportunities to pursue fundamental understanding of the stability[11–13], isomerism[14], optical[15–18], chiroptical[3], catalytic[2, 19, 20], and magnetic[21–23] properties of Au nanoparticles.

Single-atom doping has gained significant interest for its potential to design novel bi-functional heterogeneous catalysts with superior or new properties compared to the homo-gold counterparts[1, 24]. For example, it has been demonstrated that a single atom of Pd, Pt, Cd, and Hg can be successfully doped into gold nanoparticles not only to enhance the stability of the nanoparticle but also to tune the catalytic and optical properties of the nanoparticle[25–29]. It is worth noting that the reported single-atom doped/alloyed gold nanoparticles are mainly limited to heterometals from a different group of elements rather than in the same group as gold (i.e., Cu, Ag). For example, a work done by Copley et al.[10] shows that reaction between $[Au_{11}(PMePh_2)_{10}]^+$ and $[MCl(PMePh_2)]$ (M = Ag or Cu) results in the formation of a nanocluster with multiple heterometals, i.e., $[Au_9M_4Cl_4(PMePh_2)_8]^+$. This reaction is believed to occur through intermediate cations containing different numbers of metal dopants, i.e., $[Au_{11}M_2Cl_2(PMePh_2)_{10}]^{3+}$ and $[Au_{10}M_3Cl_3(PMePh_2)_9]^{2+}$[10]. Another interesting finding by Bakr and co-workers[30] is that the single Pd atom in the $Pd_1Ag_{24}$ nanocluster could be replaced by a gold atom, resulting in single gold atom-doped $Au_1Ag_{24}$. Despite many efforts, preparation of gold nanoparticles doped with a single Cu or Ag atom still remains challenging due to the similar configuration of outmost electrons ($d^{10}s^1$) of Cu and Ag as that of Au. This similarity leads to easy formation of a distribution of Cu or Ag dopants in the alloy nanoparticles[31–35].

Although the similarity in electronic structure of Ag and Cu with Au ($d^{10}s^1$) poses a major challenge for single-atom doping of gold nanoparticles, we rationalize that a single atom of Ag or Cu should easily fill into a vacancy if the latter is pre-formed within the gold nanoparticle. This method may be able to circumvent the limitation from the similar electron configuration of the same group metals. In terms of hollow gold nanoparticles, Das et al.[36] reported a centrally hollow $[Au_{24}(PPh_3)_{10}(SC_2H_4Ph)_5Cl_2]^+$ nanoparticle formed by reaction of non-hollow $[Au_{25}(PPh_3)_{10}(SC_2H_4Ph)_5Cl_2]^{2+}$ with excess triphenylphosphine ($PPh_3$). Single crystal X-ray diffraction analysis shows that the nanoparticle consists of two incomplete icosahedral $Au_{12}$ units linked by five thiolate linkages[36]. In comparison to the vertex-sharing biicosahedral $[Au_{25}(PPh_3)_{10}(SC_2H_4Ph)_5X_2]^{2+}$, the $Au_{24}$ nanoparticle lacks the central Au atom (i.e., the shared vertex atom in the biicosahedral $Au_{25}$), which exerts a major influence on the optical properties of the nanoparticle[36, 37]. This hollow structure opens up the possibility of re-filling the central vacancy of the $Au_{24}$ nanoparticle by another atom from the same group as gold. Since there is only one vacancy in the $Au_{24}$ nanoparticle, we expect that single-atom doping can be realized by using hollow $Au_{24}$ as a template. Furthermore, by subsequently hollowing the resultant single-atom alloyed nanoparticles and then re-filling with a heterometal atom, one may achieve atom-by-atom doping in a highly controlled fashion.

Herein, we report the shuttling of single metal atom(s) of Au, Ag, and Cu using the hollow $Au_{24}$ nanoparticle as a model system. Surprisingly, we discover intriguing pathways of shuttling for different metals. Instead of simple filling of the central vacancy, we find that the incoming atom squeezes the pre-existent gold atom of the nanoparticle into the hollow site to produce $M_1Au_{24}$ nanoparticles (M = Au/Ag/Cu). The obtained non-hollow $M_1Au_{24}$ nanoparticles can be further converted to $M_2Au_{23}$ nanoparticles by the hollowing-refilling strategy. The determination of the atomic structures of $Cu_1Au_{24}$ and $Ag_1Au_{24}$ nanoparticles by X-ray crystallography, together with density functional theory (DFT) simulations, provides a clear map on how the single-atom shuttling occurs in the atomically precise nanoparticles.

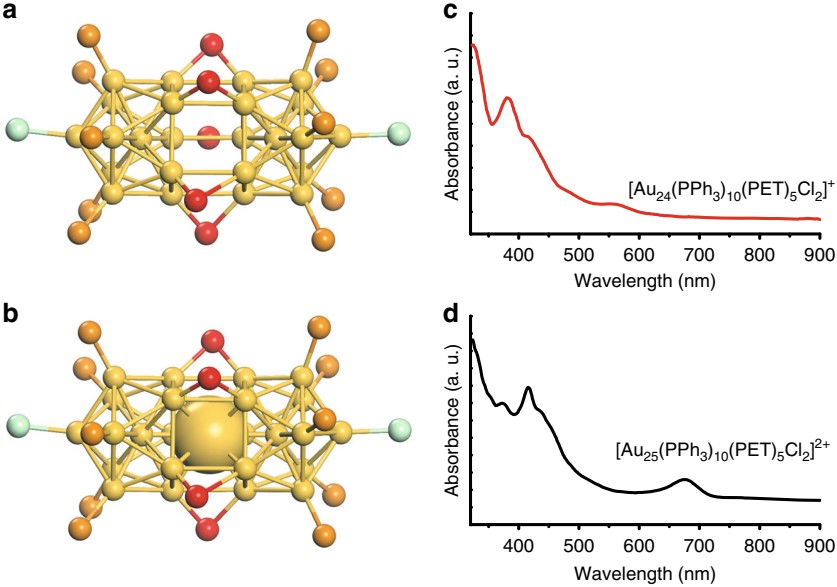

**Fig. 1** X-ray structures and UV–Vis spectra of $Au_{24}$ and $Au_{25}$ nanoclusters. X-ray structures of the hollow $Au_{24}$ rod with the central atom dislodged (**a**), and the $Au_{25}$ rod (**b**)[36, 37], the central gold atom in the $Au_{25}$ rod is shown using space-filling model for clarity. Color code: Au, yellow; P, orange; S, red; Cl, green. C and H atoms are not shown for clarity; UV–Vis spectra of the hollow $Au_{24}$ rod and the $Au_{25}$ rod are shown in **c** and **d**, respectively

## Results

**Shuttling a metal atom into a hollow nanoparticle.** The hollow $[Au_{24}(PPh_3)_{10}(SC_2H_4Ph)_5Cl_2]^+$ nanoparticle (Fig. 1a, abbreviated as Au$_{24}$ hereafter) is chosen as a model to demonstrate the filling of the central vacancy and dislodging of an atom out of the

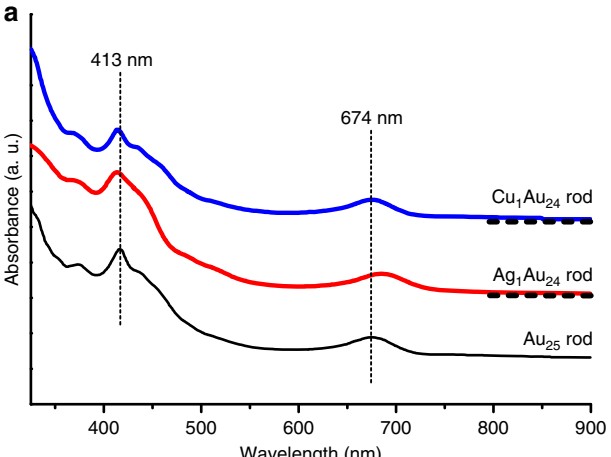

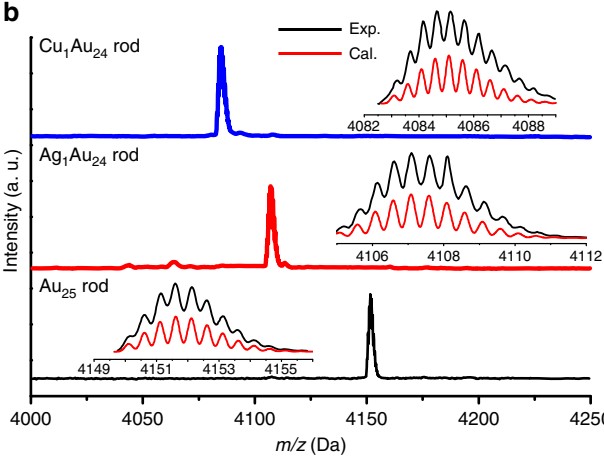

**Fig. 2** UV–Vis and ESI-MS spectra of homo-gold and doped MAu$_{24}$ nanoclusters. **a** UV–Vis spectra of homo-gold $[Au_{25}(PPh_3)_{10}(PET)_5Cl_2]^{2+}$ (black line), single Ag doped $[Ag_1Au_{24}(PPh_3)_{10}(PET)_5Cl_2]^{2+}$ (red line), and single Cu doped $[Cu_1Au_{24}(PPh_3)_{10}(PET)_5Cl_2]^{2+}$ (blue line); **b** Positive mode ESI-MS spectra of homo-gold $[Au_{25}(PPh_3)_{10}(PET)_5Cl_2]^{2+}$ (black line), single Ag doped $[Ag_1Au_{24}(PPh_3)_{10}(PET)_5Cl_2]^{2+}$ (red line), and single Cu doped $[Cu_1Au_{24}(PPh_3)_{10}(PET)_5Cl_2]^{2+}$ (blue line)

resultant 25-atom nanoparticle (Fig. 1b, abbreviated as Au$_{25}$ hereafter). The hollow Au$_{24}$ nanoparticle was made by the reaction of $[Au_{25}(PPh_3)_{10}(SR)_5Cl_2]^{2+}$ with excess PPh$_3$[36].

In the present work, we have discovered that reaction of the Au$_{24}$ (dissolved in CH$_2$Cl$_2$, Fig. 1c) with Au(I)Cl readily restores Au$_{25}$ within a few seconds, evidenced by ESI-MS analysis of the final product (Fig. 2b, black line) with a major peak of 2+ charge at $m/z = 4151.6$ Da (expected $m/z = 4151.6$ Da), also evidenced by the UV–Vis spectrum (Fig. 2a) being identical to that of Au$_{25}$ (Fig. 1d)[37]. In order to obtain single-atom doping with Ag and Cu, we further tested the Au$_{24}$ with Cu(I)Cl and Ag(I)Cl salts. Results show that addition of CuCl or AgCl to a dichloromethane solution of the Au$_{24}$ leads to a rapid (~4 s) change of the solution color from red to green, indicating the possible formation of new products doped with Cu or Ag. The UV–Vis spectrum of the Cu-doped nanoparticle is found to be similar to that of the Au$_{25}$ nanoparticle (Fig. 2a, blue line), while the Ag-doped nanoparticle exhibits a slight red shift by ~11 nm (Fig. 2a, red line). ESI-MS analysis of the doped clusters (Fig. 2b) shows that the major mass peak for Cu doping is located at $m/z = 4085.1$ (Fig. 2b, blue line), assigned to $[Cu_1Au_{24}(PPh_3)_{10}(PET)_5Cl_2]^{2+}$ (theoretical $m/z = 4085.1$ Da), and for Ag doping, the peak at $m/z = 4107.0$ (Fig. 2b, red line) corresponds to $[Ag_1Au_{24}(PPh_3)_{10}(PET)_5Cl_2]^{2+}$ (theoretical $m/z = 4107.1$ Da).

We further crystallized the products and performed X-ray crystallography to determine the sites occupied by the incoming Cu and Ag atoms in the structure of the doped nanoparticles (for details see Supplementary Figs. 1–3 and Supplementary Tables 1 and 2). Since the atomic numbers of Cu ($Z = 29$) and Ag ($Z = 47$) are considerably less than that of Au ($Z = 79$), they can be readily differentiated in the X-ray crystallographic analysis. Partial occupancy analysis was employed to find the location of Cu and Ag atoms (details are given in the Supplementary Note 2). Results show that Cu can occupy either the apex or waist positions of the rod-shaped nanoparticle (Fig. 3, right), while Ag was only found at the apex of the nanoparticle (Fig. 3, left). Interestingly, the central position of the nanoparticle is 100% occupied by gold atom in both products, rather than a Cu or Ag atom, as one would expect since the central vacancy is ready for filling.

The results of Au$_{24}$ reaction with AuCl, AgCl, and CuCl clearly demonstrate the success in single-atom doping into the gold nanoparticle. In the case of reaction with AuCl, the pathway of how the central vacancy is filled cannot be revealed, but the reactions of Au$_{24}$ with AgCl and CuCl clearly show that the copper or silver atom does not directly take the central empty position as one would initially expect, instead the Cu or Ag dopant should squeeze one surface gold atom into the central

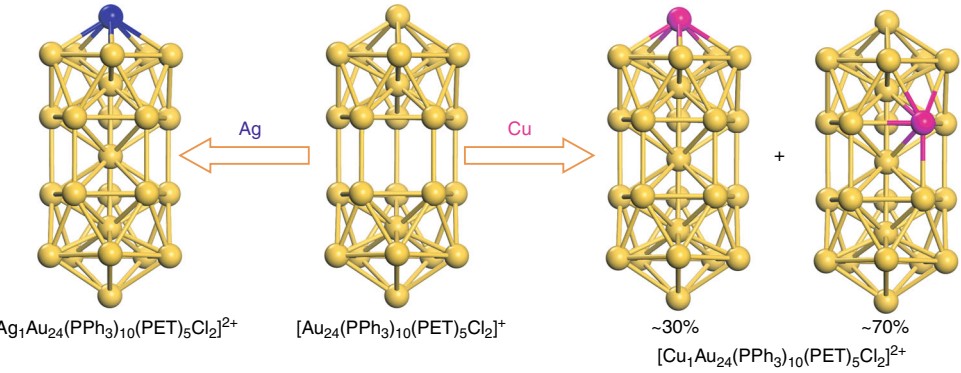

**Fig. 3** Shuttling one Ag or Cu atom into the 24-atom hollow gold nanoparticle: pathways of single Ag/Cu entering the hollow Au$_{24}$ nanoparticle. Note, Ag$_1$Au$_{24}$ and Cu$_1$Au$_{24}$ are presented using X-ray cryptographic data of this work, and Au$_{24}$ structure is adopted from ref. [36]. Color codes: Au, yellow; Ag, blue; Cu, magenta. Other non-metal atoms are not shown for clarity

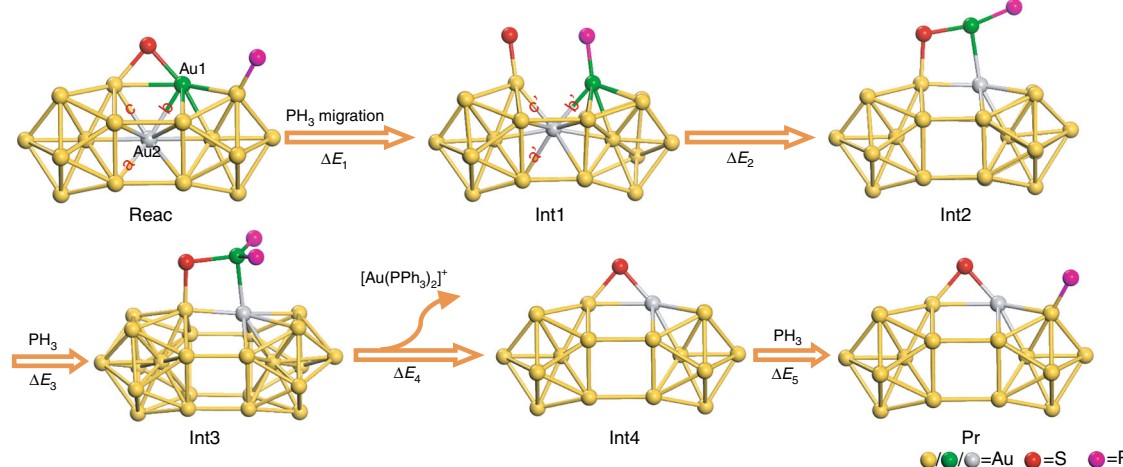

**Fig. 4** Mechanisms for the formation of hollow $Au_{24}$ cluster proposed by DFT calculations. The values of interatomic distances are $a = b = c = 2.97$, $a' = 3.23$, $b' = 2.90$, and $c' = 2.86$ Å. DFT results show $\Delta E_1 = 25.9$, $\Delta E_2 = 7.1$, $\Delta E_3 = -5.2$, $\Delta E_4 = 34.4$, and $\Delta E_5 = -30.7$ kcal/mol. Color codes: Au1, green; Au2, gray; other Au, yellow; S, red; P, magenta. Other atoms and bonds are not shown for clarity

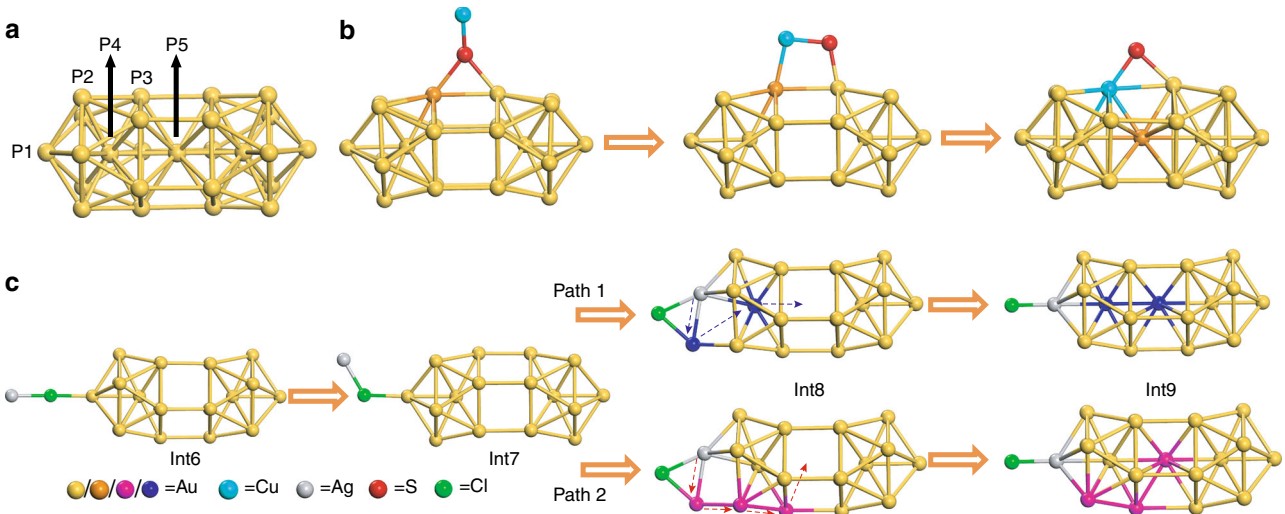

**Fig. 5** Mechanisms for the formation of doped $MAu_{24}$ clusters based on DFT calculations. **a** Designation of sites P1–P5 in the $Au_{25}$ structure. DFT-calculated mechanisms for $MAu_{24}$ (M = Cu or Ag) formation with M at **b** waist and **c** apex positions. Color code: Au, yellow; S, red; Cl, green; Cu, cyan; Ag, gray. Note, in **b**, a gold atom at the waist position that is pushed into the vacancy to form $CuAu_{24}$ is shown in orange. In **c**, to show two different pathways, i.e., paths 1 and 2, corresponding gold atoms are either presented in dark blue or magenta. Only one Cl and S of the nanoparticle is presented. Other atoms and bonds are not shown for clarity

vacancy. To map out the mechanistic details, we further carried out DFT simulations on the formation of hollow $Au_{24}$ from the $Au_{25}$ nanoparticle and the back filling of $Au_{24}$ to form $MAu_{24}$ (M = Cu or Ag).

**On the shuttling-out mechanism for the formation of hollow $Au_{24}$.** Experimentally we found that excess phosphine ligands play a key role in the formation of hollow $Au_{24}$ nanoparticle from its parent $Au_{25}$ nanoparticle, in agreement with the previous study[36]. DFT calculations were performed using $[Au_{25}(PH_3)_{10}(SH)_5Cl_2]^{2+}$ as a model of the experimental nanoparticle by simplifying $PPh_3$ to $PH_3$ and $SC_2H_4Ph$ to $SH$. Results show that adsorption of a $PH_3$ onto a gold atom located at the waist position (Au1, Fig. 4, green ball) of the nanoparticle is the most likely mechanism to initiate the reaction. A $PH_3$ of the rod via a migration process (Reac → Int1, Supplementary Movie 1) may form a bond with the Au1. Of note, the Au–$PPh_3$ bond is

flexible, which allows rapid exchange between the free and bound $PPh_3$[38].

Upon the formation of Au–$PH_3$ bond and subsequent Au–S bond breaking (Int1, Fig. 4), the gold atom at the center of the nanoparticle (Au2, Fig. 4, gray ball) dislocates toward the surface of the nanoparticle, evidenced by changes in the Au–Au atomic distances (Fig. 4). The Au1–Au2 bond distance becomes 2.90 Å ($b'$ in Fig. 4) which is considerably less than the bond distance between Au2 and gold atoms located at the lower side of the waist position ($a' = 3.23$ Å, Fig. 4). Next, the Au2 is completely pulled up to the surface of the nanoparticle (Int1 → Int2 and Supplementary Movie 2). In turn, this exposes the Au1 to $PH_3$ ligands in the reaction medium to form $Au(PH_3)_2$ on the surface of the nanoparticle (Int2 → Int3). The $Au(PH_3)_2^+$ moiety eventually detaches from the nanoparticle to result in the $[Au_{24}(PPh_3)_9(SR)_5Cl_2]^+$ nanoparticle (Int3 → Int4). The generation of $Au(PPh_3)_2^+$ ion is indeed experimentally confirmed by ESI-MS (Supplementary Fig. 4). Finally, the as-formed

$[Au_{24}(PPh_3)_9(SR)_5Cl_2]^+$ nanoparticle reacts with a $PH_3$ to result in the hollow $[Au_{24}(PPh_3)_{10}(SR)_5Cl_2]^+$ nanoparticle (Int4 → Pr).

**On the shuttling-in mechanism to form $CuAu_{24}$ and $AgAu_{24}$.** The $MAu_{24}$ nanoparticle has five non-equivalent types of metal positions (P1–P5 as indicated in Fig. 5a). Geometry optimizations

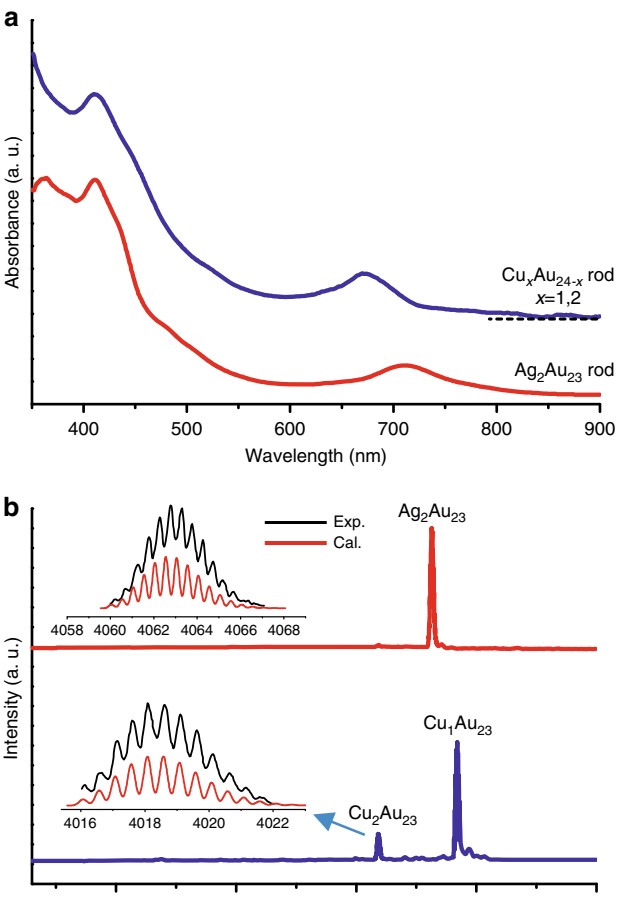

**Fig. 6** UV–Vis and ESI-TOF-MS spectra of the secondary shuttling products. **a** UV–Vis spectra of $[Cu_xAu_{25-x}(PPh_3)_{10}(PET)_5Cl_2]^{2+}$ ($x=1,2$; blue line) and $[Ag_2Au_{23}(PPh_3)_{10}(PET)_5Cl_2]^{2+}$ (red line), and; **b** Positive mode ESI-MS spectra of $[Cu_xAu_{25-x}(PPh_3)_{10}(PET)_5Cl_2]^{2+}$ ($x=1,2$; blue line) and $[Ag_2Au_{23}(PPh_3)_{10}(PET)_5Cl_2]^{2+}$ (red line)

of $MAu_{24}$ with M located at the different positions show that both Cu and Ag energetically disfavor to occupy positions of P2 and P4 (Supplementary Tables 3 and 4), in good agreement with the X-ray crystallography analysis. However, relative energetics of the nanoparticles with M at positions P1, P3, and P5 are not in line with the experimental results. DFT results show that Cu prefers to occupy the sites in the order of P1 ≈ P5 > P3, while for the case of Ag, the order is P5 > P3 > P1 (Supplementary Tables 3 and 4). For completeness, the Grimme-D2[39] and the exchange hole dipole moment (XDM)[40, 41] methods were used to incorporate the van der Waals (vdW) interactions into the systems. As Supplementary Tables 3 and 4 show, DFT-D2 and DFT-XDM calculations yield nearly the same results as DFT does. The X-ray crystallography analysis indicates that Ag prefers to locate at site P1 and Cu at P3 (Fig. 3); therefore, our calculations reveal that in addition to the relative stability of the nanoparticles based on their energetics, other factors such as reaction kinetics and entropy effects (10 P3 sites vs two P1 sites) also play significant roles in the formation of $MAu_{24}$.

We next considered whether the location of incoming M (Cu or Ag) is dictated by the initial interaction of $M^+$ with the capping ligands of the nanoparticle (-SR and Cl–Au, Supplementary Fig. 5). Interaction energy of $Ag^+$ and $Cu^+$ with Cl–Au is found to be 9.6 and 8.0 kcal mol$^{-1}$, respectively, more favorable than those with—SR. These results show that the interaction of Cl–Au with Ag is stronger than with Cu. This may indicate the single-atom transfer and its possible location is determined by the interaction of the M (Cu or Ag) with capping ligands of the nanoparticle, in agreement with our experimental trend. In addition, compared to Cu, the larger vdW radius of Ag also prevents the silver atoms from interacting efficiently with -S- due to steric hindrance. Of note, the protecting ligands of the nanoparticle make the particle surface considerably packed, which causes high spatial hindrance for $Ag^+$ to pass through and approach the -SR group (Supplementary Fig. 6). However, Cu has a smaller vdW radius and can interact with surface thiolate ligand, which eventually pushes a gold atom at the waist position into the vacancy at the center of $Au_{24}$ to form $CuAu_{24}$ (Fig. 5b).

Further, we consider possible mechanisms to form $AgAu_{24}$ with Ag located at the apical site of the nanoparticle. An $Ag^+$ may interact with the Cl atom at the apex of the nanoparticle (Int6, Fig. 5c), which eventually moves to interact with three gold atoms located at the apical as well as the end positions of the nanoparticle (Int6 → Int7 → Int8, Fig. 5c). There are two possible pathways for the Ag at this position to locate at the apical site by squeezing a gold atom at sites of icosahedral center (Fig. 5c, Path 1, shown by blue arrows, Supplementary Movie 3) and waist

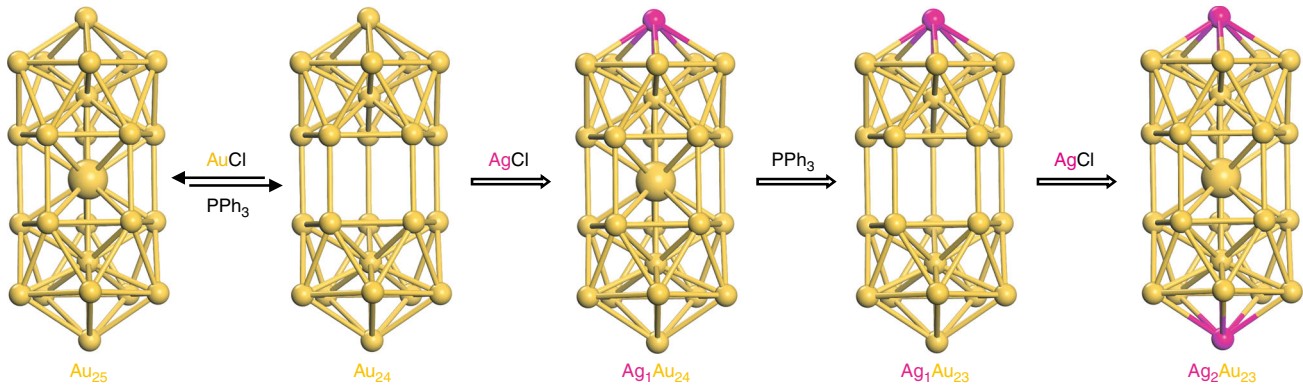

**Fig. 7** Proposed mechanism of injecting two Ag atoms into the nanoparticle via the hollowing-refilling sequence: step 1, using $PPh_3$ to make a hole in the solid $Au_{25}$ nanocluster; step 2, using AgCl to refill this hole and produce $Ag_1Au_{24}$; step 3, continue using $PPh_3$ to make a hole in the $Ag_1Au_{24}$ nanocluster and form the hollow $Ag_1Au_{23}$ nanocluster; step 4, using AgCl to refill the hole and yield the $Ag_2Au_{23}$ nanocluster. Color code: Au, yellow; Ag, magenta

(Fig. 5c, Path 2, shown by red arrows, Supplementary Movie 4). Our calculations using the nudged elastic band (NEB) approach[42] show barrier energy of pathway 2 is 19.8 kcal mol$^{-1}$ lower than that for pathway 1. This result indicates metal mobility is most likely to happen through the surface of the nanoparticle rather than the core of the icosahedron, in agreement with the mechanism for the $Au_{24}$ formation.

**Shuttling a second heteroatom into the nanoparticle**. To shuttle a second heteroatom into the nanoparticle, the $Cu_1Au_{24}$ and $Ag_1Au_{24}$ nanoparticles were, respectively, used as the starting material. Reaction of the starting material with $PPh_3$ at 40 °C produced hollow nanoparticles. As shown in Supplementary Fig. 7, the complete disappearance of the 700 nm peak indicates that all the $M_{25}$ nanoparticles have been converted to hollow $M_{24}$. The second step is to fill the hollow structure with heterometal atom by adding CuCl or AgCl salts to the solution. The color of the solution changed immediately from red to green. As shown in Fig. 6a, compared with $Au_{25}$, the copper-doped product has a similar UV–Vis spectrum as that of $Au_{24}$, however, the silver-doped nanoparticle shifted from ~685 nm ($Ag_1Au_{24}$) to ~712 nm. In the ESI-MS spectra (Fig. 6b), the $Ag_2Au_{23}$ nanoparticle with +2 charge was found ($m/z = 4062.8$ Da, theoretical $m/z = 4062.6$ Da), which implies a step-by-step doping of silver to the two apex sites (Fig. 7). For Cu doping, the product comprises $Cu_1Au_{24}$ (major, $m/z = 4085.1$, theoretical $m/z = 4085.1$) and $Cu_2Au_{23}$ (less, $m/z = 4018.1$ Da, theoretical $m/z = 4018.1$ Da). To explain why $Cu_1Au_{24}$ is the major product, we note that the starting $Cu_1Au_{24}$ material is a mixture of apex- and waist-doped nanoparticles, and the strong binding of $PPh_3$ to Cu should cause the dislodging of waist Cu atom in the $Cu_1Au_{24}$ (70% population, see Fig. 3 above) to produce the hollow $Au_{24}$ nanoparticle, and then reaction of $Au_{24}$ with CuCl produces $Cu_1Au_{24}$, while the apex-doped $Cu_1Au_{24}$ (30% population, Fig. 3) produces $Cu_1Au_{23}$ and its reaction with CuCl gives rise to $Cu_2Au_{23}$, hence a minor component in the product.

## Discussion

In summary, we have successfully implemented the single-metal atom shuttling into an atomically precise metal nanoparticle and mapped out the mechanism of the conversion between the $Au_{24}$ and the $Au_{25}$ nanoparticles. Our results provide a clear map of how single metal atom transfer occurs between two atomically precise nanoparticles. Based on the experimental and theoretical results, the driven force of single-atom transfer is caused by the ligand, i.e., the free $PPh_3$ for the shuttling-out process, and the surface -Cl and -SR ligands for the shuttling-in process. The stronger binding between Ag and -Cl compared with Ag–SR leads to the exclusive Ag atom doping at the apex of the nanoparticle, while the similar energy of Cu–Cl and Cu–SR leads to the Cu atom doping into both the apex and waist positions. This work provides fundamental understanding of how to shuttle a single atom in and out of metal nanoparticles by a chemical method. The ligand-induced single-atom shuttling process also provides a strategy for controlling the doping position and the doping number of heteroatoms in alloy nanoparticles.

## Methods

**Materials**. Unless specified, reagents were purchased from ACROS Organics or Sigma-Aldrich and used without further purification. Tetrachloroauric(III) acid ($HAuCl_4 \cdot 3H_2O$, >99.99% metals basis), CuCl (99%), AgCl (99%), AuCl (99%), $NaSbF_6$ (>99%), $PPh_3$ (>99%), and $NaBH_4$ (>98%) were received from ACROS Organic. Ethanol (HPLC grade, ≥99.9%), methanol (HPLC grade, ≥99.9%), and methylene chloride (HPLC grade, ≥99.9%) were from Sigma-Aldrich. UV–Vis absorption spectra were obtained using an Agilent 8453 instrument, and solution samples were prepared using DCM as the solvent. ESI-MS was recorded using a Waters Q-TOF mass spectrometer equipped with Z-spray source. The source

temperature was kept at 70 °C. The sample was directly infused into the chamber at 5 μL min$^{-1}$. The spray voltage was kept at 2.20 kV and the cone voltage at 60 V. $[Au_{24}(PPh_3)_{10}(SR)_5Cl_2]^+$ and $[Au_{25}(PPh_3)_{10}(SR)_5Cl_2]^{2+}$ were synthesized according to the literature method[34, 35] (for details see Supplementary Note 1).

**$[M_1Au_{24}(PPh_3)(SR)_5Cl_2](SbF_6)_2$ (M = Au/Ag/Cu)**. The $[Au_{24}(PPh_3)(SR)_5Cl_2]$Cl nanocluster (~2 mg) was dissolved in $CH_2Cl_2$, then ~1 mg MCl salt (M = Au/Ag/Cu) was added into the solution, respectively. After shaking for a few seconds, the solution color rapidly changed from red to green. Then, the solution was centrifuged to remove the exceed salt (solid), and the solution was then dried under $N_2$. To exchange for the anion, the obtained nanoclusters were dissolved in EtOH, then a right amount of $NaSbF_6$ was added into the solution. The precipitate was collected after centrifugation, followed by crystallization in dichloromethane/pentane.

**$[M_2Au_{23}(PPh_3)(SR)_5Cl_2](SbF_6)_2$ (M = Ag/Cu)**. Approximately 5 mg of $[M_1Au_{24}(PPh_3)(SR)_5Cl_2]Cl_2$ nanoclusters (M = Ag/Cu) was dissolved in 2 mL $CH_2Cl_2$ solution, followed by adding 1 g of $PPh_3$. The reaction was allowed to proceed overnight at 40 °C. Then, 10 mL of hexane was added to remove the excess $PPh_3$. Then, 1 mg of MCl salt (M = Au/Ag/Cu) was added into the solution, respectively. After shaking for a few seconds, the solution color changed from red to green. Then, the solution was centrifuged to remove the excess salt, and the solution was dried under $N_2$.

**X-ray crystallography**. The data collections for single crystal X-ray diffraction was carried out on a Bruker Smart APEX II CCD diffractometer, using a Cu–$K_\alpha$ radiation ($\lambda = 1.54178$ Å). Data reductions and absorption corrections were performed using the SAINT and SADABS programs[43], respectively. The structure was solved by direct methods and refined with full-matrix least squares on $F^2$ using the SHELXTL software package[44]. All non-hydrogen atoms were refined anisotropically, and all the hydrogen atoms were set in geometrically calculated positions and refined isotropically using a riding model. X-ray diffraction data refinement involving partial occupancy was used to locate the heteroatom atom.

**Computational details**. DFT calculations were carried out using the Quantum Espresso package[45]. The Projector Augmented-Wave (PAW) method was applied to describe the interaction between the electrons and nuclei[46]. The Perdew–Burke–Ernzerhof (PBE) form of the generalized gradient approximation was employed for electron exchange and correlation[47]. The gold cluster was placed at the center of a cubic box of 30.0 Å × 30.0 Å × 30.0 Å. The kinetic energy cutoff was chosen to be 450 eV and integration in the reciprocal space was carried out at the Γ k-point of the Brillouin zone. The NEB approach was used to find minimum energy path of transitions[42].

**Data availability**. The X-ray crystallographic coordinates for structures reported in this work (see Supplementary Tables 1, 2, and Supplementary Note 2) have been deposited at the Cambridge Crystallographic Data Centre (CCDC), under deposition numbers CCDC 1562010 and CCDC 1561987. These data can be obtained free of charge from The Cambridge Crystallographic Data Centre via www.ccdc.cam.ac.uk/data_request/cif.

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

## Acknowledgements

S.W. acknowledges the scholarship support from the China Scholarship Council. M.Z. thanks the financial support from NSFC 21372006 &U1532141, the Ministry of Education, the Education Department of Anhui Province, 211 Project of Anhui University. This work was also supported in part by National Science Foundation through NSF Grant No. CHE-1223988 (H.J.K.). R.J. acknowledges financial support from the U.S. National Science Foundation (DMREF-0903225).

## Author contributions

S.W. and H.A. contributed equally to this work. S.W. synthesized the samples and carried out the experimental tests. H.A. and H.J.K. performed the DFT calculations. C.L., T.-Y.L and N.L.R. solved the crystal structures. S.W., M.Z., and R.J. designed the study. S.W., H.A., and R.J. wrote the manuscript. All authors discussed the results and commented on the manuscript.

## Additional information

**Competing interests:** The authors declare no competing financial interests.

