## [Peer Review File · Nature Communications]

Reviewers' comments:

Reviewer #1 (Remarks to the Author):

Study on alloy cluster is one of the most interest topic in the cluster chemistry, because replacement of Au with other element can change their property. However silver and copper has similar electron configuration, this lead the mixture of different number of doped clusters in as-prepared sample. The author demonstrated precise doping of Ag and Cu atom into phosphine/halogen-protected Au₂₅ cluster by chemical reaction of Au₂₄ and MCl salts. They also found out the mechanism by calculations. The detailed discussion was very interesting and I really enjoyed reading manuscript. Thus, I recommend this manuscript as Nature Communication paper after the following revisions.

I have a few comments on your manuscript.

- 1) In Abstract. The authors described "While doping gold nanoparticles with Ag or Cu typically results in alloys with a nonspecific number (i.e. a distribution) of Ag or Cu dopants". However, as you see the literature (D. Michael P. Mingos et. al., J. Chem. Soc., Dalton Trans., 1996, 491-500), the number of doped Ag and Cu was controlled. The authors should mention about this reported literature.
- 2) Page 2. In the sentence of "Results show that addition of CuCl ...red to green, indicating the formation of new products doped with Cu or Ag", because changing of color is not an evidence for confirming replacement of foreign atoms, the authors should modify the word "indicating".
- 3) Page 4. The order of preferential site of P1 = P5 > P3 for Cu (or P5 > P3 > P1 for Ag) is difficult to understand. Is it an order of energy for producing cluster? If I see this order, I little confused because, for Cu doping, priority of P3 is smaller than that of P5 (P5 > P3; for Cu).

Reviewer #2 (Remarks to the Author):

This manuscript by Jin et al. reports an important study towards understanding the mechanism of metal atom doping in monolayer protected clusters. They use rod-shaped [Au₂₄(PPh₃)₁₀SC₂H₄Ph)₅Cl₂]²⁺ and the mechanism of single metal atom shuttling into and out of this cluster is studied by its reactions with metal complexes such as AuCl, AgCl and CuCl. The authors have found important correlations between the relative affinities of dopant metal ions with ligands and the sites they occupy. Understanding mechanisms of metal atom doping into these clusters are poorly known and this is one of the earliest such studies. However, it is important to remember that there is a role of valency of the metal in the substitution the effect of which is subtle.

I note that there exists at least one related study by Bakr et al. (Distinct metal-exchange pathways of doped Ag₂₅ nanoclusters *Nanoscale*, 2016, 8, 17333–17339) on the mechanistic pathways of metal exchange in M₂₅(SR)₁₈ systems. Through mass spectrometry, they have identified the intermediates of the reaction and also they have rationalized the reaction mechanism with support from the superatom concept, correlating the overall charge states, probable sites of metal atoms and the shell closing free electron count. However, the rod-shaped Au₂₄/Au₂₅ systems reported in this manuscript are not superatomic and hence such correlations may not be possible. In the present study by Jin et al., though they have resolved the single crystal x-ray structures of the final products and the starting clusters, mechanistic insights come mostly from the computational results and they do not have a direct experimental evidence for the intermediates and dynamics of the events, unlike the results of Bakr. This work has to be cited in the manuscript.

In the section, "On the shuttling-out mechanism for the formation hollow Au₂₄", the authors wrote that "Experimentally we found that excess phosphine ligands play a key role in the formation of hollow Au₂₄ nanoparticle from its parent Au₂₅ nanoparticle". This result is in fact from a reported paper by

the same author (Ref.33). In the current work also, the authors primarily depend on this reaction. The only new contribution from this section is computational results on the mechanism of this reaction. No new experimental evidence on its mechanism is presented in this section.

In the introduction, the authors mentioned about tunneling, however it is not clear as to what does it mean, though the authors refer to it as "simple filling of the central vacancy". In such clusters containing multiple metal atom sites, it is quite arguable/understandable that initial events of doping would involve binding to the outer sites, followed by dynamic rearrangement to thermodynamically preferred (inner/outer) sites. Do the authors mean "tunneling" as direct filling of the central vacancy without involving outer sites in the initial steps of reaction? However, such a reaction is highly unlikely as it is impossible for a metal atom to reach the central cavity without disrupting the outer metal-ligand bonding network. If any such "tunneling" is reported in the literature, the authors should cite those papers.

In this study, the authors have attributed a significant role of ligands such as Cl⁻, PPh₃, in deciding the dopant atom locations in the products. Relative affinities of Ag⁺/Cu⁺ to Cl⁻ or sulfur atoms (of ligands) will be available from the literature from which a guess of probable initial metal atom binding sites can be made even without calculations. Also, authors may mention how the affinity of Ag⁺/Cu⁺ to a free Cl⁻ or free -SR ligand differs from that between Ag⁺/Cu⁺ towards those sites in the cluster (as shown in Supplementary Figure 4).

Authors attribute the single atom doping to the higher affinity of Ag⁺ to Cl⁻ at the apex (P1) sites which is supported by the experimentally observed structure. However, calculations presented in Supplementary Table 1 is not at all in agreement in this. Hence, other possible factors governing the stability of such alloy systems are to be considered. For example, a discussion of the bond length and bond strength in the doped and undoped clusters may be necessary.

There are two identical apex or P1 sites with Cl⁻ as the ligand. Hence, why is it not possible to obtain [Ag₂Au₂₃(SR)₅(PPh₃)₁₀Cl₂]²⁺ by direct reaction of [Au₂₄(SR)₅(PPh₃)₁₀Cl₂]²⁺ with Ag⁺? In other words, a simple replacement of the apex Au atoms in [Au₂₄(SR)₅(PPh₃)₁₀Cl₂]²⁺ by added Ag⁺ is also likely. Authors should discuss this possibility in their discussion of reaction mechanisms. Also, is it not possible to incorporate more than 2 Ag atoms (or Cu atoms) in these clusters through multiple steps of hollowing-filling strategy? I see that multiple Ag and Cu atom incorporation is indeed possible in icosahedral (not rod-shaped), all-thiolate protected systems which also contain M₁₃ icosahedra. Hence, these experiments will give valuable insights on the role of the ligands on the extent of metal atom doping.

There is ambiguity in the labelling of metal atom locations P3 and P4. In Figure 5 of the main paper, P3 is the waist positions while in Supplementary Figures 1 and 2, P3 is shown as the central position. This has to be clarified otherwise the whole discussion about the site preference becomes unclear.

There is an error in the first paragraph of the section entitled, On the shuttling-in mechanism for the formation of MAu₂₄ (M=Cu or Ag). Figure 2 is wrongly referred in the sentence "The X ray crystallography analysis indicates that.....(Fig 2)". In fact, Fig 2 does not contain any structures. It is not clear what the experimentally observed crystal structure in this work is. Are the structures shown in Figure 3 refer to the crystal structures found in this work?

In the captions of figures showing the structures (except that of Fig 1), it is not clearly mentioned whether they are crystal structures, resolved in this work, or a schematic made from reported coordinates. This has to be explicitly mentioned in the captions in main paper and in the SI.

In Figure 4, it is not explicitly mentioned whether the mechanism shown is from the DFT calculations performed in this work or not. In Figure 5, this is mentioned explicitly.

It is not clear whether Supplementary Figures 1 and 2 show the experimentally resolved structures of $\text{Cu}_1\text{Au}_{24}$ and $\text{Ag}_1\text{Au}_{24}$ or just a general schematic showing the distinct metal atom locations.

In Supplementary Figures 1 and 2, it is better to mention the positions of Cu and Ag. Also it has to be mentioned in the captions that (i) only metal atoms are shown, (ii) P1, P2, etc., are various symmetry unique metal atom positions. Also, the reader has no idea (i) what are the structures shown in the right of these figures conveying and (ii) why two structures are shown.

In Supplementary Figure 4, details of the energy differences to be explained in the caption. Also the color codes to be mentioned more clearly.

In Supplementary Figure 5, color codes of atoms have to be mentioned explicitly. Also, it is advisable to mention that the figure shows a lateral/top view or something like that. It is mentioned in the caption that S atoms are green, however only one green sphere is shown, though there are more than five thiolate ligands.

Reviewer #3 (Remarks to the Author):

In this manuscript, the authors have investigated atomically precisely doped metal nanoclusters and found that the Ag dopant goes to the apex site of the Au_{24} rod while the Cu doping into both the apex and waist positions. Currently, the controlled preparation of doped nanomaterials is unusual. This paper provides valuable insights into the controlled doping both in position and number of heteroatoms in alloy nanoparticles, so will be of some interest to the on-going research of alloy nanoparticles. However, the methods employed in this study is similar to their previous article published in JACS 137, 4018 (2015), which may be qualified as a follow-up of earlier results on the single-atom doping into gold nanoparticles and lack of novelty to the readers. This difference of the Ag and Cu doping has been explained by the binding between the dopant and $-\text{Cl}$ ($-\text{SR}$). I think the authors should also explain their difference with the Cd doping, for which the Cd is in the center of the 13-atom icosahedral core. Furthermore, although the alloying is often used to improve the properties of bulk materials, it is far more difficult to make this kind of doping in a nanostructured material due to the so-called "self-purification effect", leading to the expulsion of impurities. The authors should explain the reason why the alloy nanoclusters can be stable against the self-purification effect. The third question is that the transition states in Figs. 4 and 5 have been checked by the frequency calculations.

In summary I think this work does not meet the criteria of novelty or urgency that apply to Nature Commun.

Manuscript number: NCOMMS-17-04253

Manuscript Type: Article

Title: "Shuttling Single Metal Atom into and out of a Metal Nanoparticle"

We thank all the reviewers for their helpful comments, which improve our manuscript. Our point-by-point response is shown in blue, and revisions to the manuscript are in red.

Response to Reviewer 1: see page 2

Response to Reviewer 2: see pages 3-6

Response to Reviewer 3: see pages 7-8

Reviewer #1 (Remarks to the Author):

Study on alloy cluster is one of the most interest topic in the cluster chemistry, because replacement of Au with other element can change their property. However silver and copper have similar electron configuration, this lead the mixture of different number of doped clusters in as-prepared sample. The author demonstrated precise doping of Ag and Cu atom into phosphine/halogen-protected Au₂₅ cluster by chemical reaction of Au₂₄ and MCl salts. They also found out the mechanism by calculations. The detailed discussion was very interesting and I really enjoyed reading manuscript. Thus, I recommend this manuscript as Nature Communication paper after the following revisions.

I have a few comments on your manuscript.

1) In Abstract. The authors described "While doping gold nanoparticles with Ag or Cu typically results in alloys with a nonspecific number (i.e. a distribution) of Ag or Cu dopants". However, as you see the literature (D. Michael P. Mingos et. al., J. Chem. Soc., Dalton Trans., 1996, 491-500), the number of doped Ag and Cu was controlled. The authors should mention about this reported literature.

Response: We thank the reviewer for bringing this ref to our attention. We have added it as new ref. 10 in the revised manuscript.

Revision:

"Copley et al. previously reported a reaction between $[\text{Au}_{11}(\text{PMePh}_2)_{10}]^+$ and $[\text{MCl}(\text{PMePh}_2)]$ (M = Ag or Cu) which resulted in the formation of a nanocluster with multiple heterometals, i.e., $[\text{Au}_9\text{M}_4\text{Cl}_4(\text{PMePh}_2)_8]^+$.¹⁰ This reaction was believed to occur through intermediate cations containing different numbers of metal dopants, i.e., $[\text{Au}_{11}\text{M}_2\text{Cl}_2(\text{PMePh}_2)_{10}]^{3+}$ and $[\text{Au}_{10}\text{M}_3\text{Cl}_3(\text{PMePh}_2)_9]^{2+}$."¹⁰

2) Page 2. In the sentence of "Results show that addition of CuCl ...red to green, indicating the formation of new products doped with Cu or Ag", because changing of color is not an evidence for confirming replacement of foreign atoms, the authors should modify the word "indicating".

Response: We have changed the sentence:

"Results show that addition of CuCl or AgCl to a dichloromethane solution of Au₂₄ leads to a rapid (within ~4 sec) change of the solution color from red to green, indicating the possible formation of new products doped with Cu or Ag."

3) Page 4. The order of preferential site of P1 = P5 > P3 for Cu (or P5 > P3 > P1 for Ag) is difficult to understand. Is it an order of energy for producing cluster? If I see this order, I little confused because, for Cu doping, priority of P3 is smaller than that of P5 (P5 > P3; for Cu).

Response: These orders (i.e., P1 = P5 > P3 for Cu and P5 > P3 > P1 for Ag) show preferential sites to be occupied by the dopants from thermodynamic point of view. These results are obtained with respect to the relative energy of the nanoparticle with a dopant at different sites. For example, the DFT results of Table 4 in Supplementary Note show that Cu prefers to occupy site P5 over P3 by 2.29 – 0.22 = 2.07 kcal/mol. Therefore, as the reviewer inferred, the priority of P3 for Cu is smaller than that of P5 (P5 > P3). A more positive energy for a site in Tables 3 and 4 of Supplementary Note indicates that such a site is energetically less probable to be occupied by a dopant. To address the reviewer's comment, we have added the following statement in Tables S3 and S4:

"A more positive energy for a position indicates that such a site is energetically less probable to be occupied by a Cu."

Reviewer #2 (Remarks to the Author):

This manuscript by Jin et al. reports an important study towards understanding the mechanism of metal atom doping in monolayer protected clusters. They use rod-shaped $[\text{Au}_{24}(\text{PPh}_3)_{10} \text{SC}_2\text{H}_4\text{Ph})_5\text{Cl}_2]^{2+}$ and the mechanism of single metal atom shuttling into and out of this cluster is studied by its reactions with metal complexes such as AuCl, AgCl and CuCl. The authors have found important correlations between the relative affinities of dopant metal ions with ligands and the sites they occupy. Understanding mechanisms of metal atom doping into these clusters are poorly known and this is one of the earliest such studies. However, it is important to remember that there is a role of valency of the metal in the substitution the effect of which is subtle.

I note that there exists at least one related study by Bakr et al. (Distinct metal-exchange pathways of doped Ag₂₅ nanoclusters *Nanoscale*, 2016, 8, 17333–17339) on the mechanistic pathways of metal exchange in M₂₅(SR)₁₈ systems. Through mass spectrometry, they have identified the intermediates of the reaction and also they have rationalized the reaction mechanism with support from the superatom concept, correlating the overall charge states, probable sites of metal atoms and the shell closing free electron count. However, the rod-shaped Au₂₄/Au₂₅ systems reported in this manuscript are not superatomic and hence such correlations may not be possible. In the present study by Jin et al., though they have resolved the single crystal x-ray structures of the final products and the starting clusters, mechanistic insights come mostly from the computational results and they do not have a direct experimental evidence for the intermediates and dynamics of the events, unlike the results of Bakr. This work has to be cited in the manuscript.

Response: We thank the reviewer for the comments. The work by Bakr et al. has been included in the revised manuscript as Ref 18. An important finding of the work by Bakr et al. is that their method of metal exchange initially resulted in a distribution of Au_xAg_{25-x} (x=1-8), in which the unstable Au_xAg_{25-x} (x=2-8) nanoclusters gradually decomposed and only Au₁Ag₂₄ was detected in their crystallographic analysis. A key feature of our work here is 100% formation yield of the rod-like Ag₂Au₂₃ or Ag₁Au₂₄, indicating our method (hollowing-refilling) is robust for single doping of metal nanoparticles.

In the section, "On the shuttling-out mechanism for the formation hollow Au₂₄", the authors wrote that "Experimentally we found that excess phosphine ligands play a key role in the formation of hollow Au₂₄ nanoparticle from its parent Au₂₅ nanoparticle". This result is in fact from a reported paper by the same author (Ref.33). In the current work also, the authors primarily depend on this reaction. The only new contribution from this section is computational results on the mechanism of this reaction. No new experimental evidence on its mechanism is presented in this section.

Response: Yes, the synthesis of Au₂₄ with excess PPh₃ is the same as in ref 33 (now re-numbered as ref 35 and indicated in the sentence). The focus of the present manuscript is to utilize the hollow Au₂₄ to do the shuttling-in and -out chemistry for controlling the doping process, which is of vast importance.

According to our computational results on the shuttling-out mechanism, we inferred that AgAu₂₄ in the presence of excess phosphine ligands would result in AgAu₂₃ with a central vacancy. In contrast, excess phosphine ligands would convert CuAu₂₄ to Au₂₄ or CuAu₂₃. These computational conclusions were further confirmed with our synthesis of Ag₂Au₂₃, CuAu₂₄ and Cu₂Au₂₃ by shuttling a second heteroatom into the particles. The combined effort of experiment and computation has provided insight into the shuttling-out mechanism.

In the introduction, the authors mentioned about tunneling, however it is not clear as to what does it mean, though the authors refer to it as "simple filling of the central vacancy". In such clusters containing multiple metal atom sites, it is quite arguable/understandable

that initial events of doping would involve binding to the outer sites, followed by dynamic rearrangement to thermodynamically preferred (inner/outer) sites. Do the authors mean “tunneling” as direct filling of the central vacancy without involving outer sites in the initial steps of reaction? However, such a reaction is highly unlikely as it is impossible for a metal atom to reach the central cavity without disrupting the outer metal-ligand bonding network. If any such “tunneling” is reported in the literature, the authors should cite those papers.

Response: We mentioned ‘tunneling’ as one of the possible scenarios in the Intro since atoms can show quantum mechanical behavior (wavefunction, superposition, etc). Even the large (~1 nm) C₆₀ molecules—which are normally viewed as ‘classical’ objects—can show quantum mechanical wave interference behavior in double-slit experiments (Nature 1999, 401, 680). While atoms are usually viewed as ‘classical’ (i.e. with definite positions), they can smear themselves into waves and do superposition, as electrons typically do. It is true that no ‘tunneling’ has been reported in doping of NPs; but, when and how atoms can exist in several positions at once, how atoms switch places, etc, remain an interesting thing and chemists have not gone that far yet.

In this study, the authors have attributed a significant role of ligands such as Cl⁻, PPh₃, in deciding the dopant atom locations in the products. Relative affinities of Ag⁺/Cu⁺ to Cl⁻ or sulfur atoms (of ligands) will be available from the literature from which a guess of probable initial metal atom binding sites can be made even without calculations. Also, authors may mention how the affinity of Ag⁺/Cu⁺ to a free Cl⁻ or free –SR ligand differs from that between Ag⁺/Cu⁺ towards those sites in the cluster (as shown in Supplementary Figure 4).

Response: The literature bond energies of M-Cl and M-S should be cautiously used because those values were from different compounds rather than from the cluster state. The results on the relative affinity of Ag⁺/Cu⁺ to a free Cl⁻ or free –SR ligand in our system are now given in the caption of revised Supplementary Figure 4.

Revision:

“For comparison, the relative affinity of Ag⁺/Cu⁺ to a free Cl⁻ or –SH⁻ in the gas phase is calculated. DFT results show that interaction energy of both Ag⁺ and Cu⁺ with a –SH⁻ is about 15 kcal/mol more favorable than that with a free Cl⁻ ion. These results indicate that the attraction of –Cl and –SR ligands with respect to Ag⁺/Cu⁺ differentiates if they are coordinated with the surface gold atoms.”

Authors attribute the single atom doping to the higher affinity of Ag⁺ to Cl⁻ at the apex (P1) sites which is supported by the experimentally observed structure. However, calculations presented in Supplementary Table 1 is not at all in agreement in this. Hence, other possible factors governing the stability of such alloy systems are to be considered. For example, a discussion of the bond length and bond strength in the doped and undoped clusters may be necessary.

Response: We agree with the reviewer that the supplementary table is not in agreement with the experimental results. Note that the computational results in the table presents the order of preferential site from energetics point view. This discrepancy is already mentioned in the manuscript and its possible origin is referred to other factors such as reaction kinetics, entropy effects, initial interaction of dopants with ligands, and steric effects of ligands that avoid efficient interactions of dopants with the nanoparticle’s surface.

There are two identical apex or P1 sites with Cl⁻ as the ligand. Hence, why is it not possible to obtain [Ag₂Au₂₃(SR)₅(PPh₃)₁₀Cl₂]²⁺ by direct reaction of [Au₂₄(SR)₅(PPh₃)₁₀Cl₂]²⁺ with Ag⁺? In other words, a simple replacement of the apex Au atoms in [Au₂₄(SR)₅(PPh₃)₁₀Cl₂]²⁺ by added Ag⁺ is also likely. Authors should discuss this possibility in their discussion of reaction mechanisms. Also, is it not possible to incorporate more than 2 Ag atoms (or Cu atoms) in these clusters through multiple steps of hollowing-filling strategy? I see that multiple Ag and Cu atom incorporation is indeed possible in

icosahedral (not rod-shaped), all-thiolate protected systems which also contain M13 icosahedra. Hence, these experiments will give valuable insights on the role of the ligands on the extent of metal atom doping.

Response: We have done the experiments. The formation of $\text{Ag}_2\text{Au}_{23}$ by direct reaction of Au_{24} with Ag^+ is not achievable. This is mainly due to the fact that upon the reaction of one Ag^+ with Au_{24} , the AgAu_{24} is formed. The later nanoparticle does not have any central vacancy; therefore, it is not possible to accommodate another Ag^+ . However, if the direct reaction takes place in the presence of excess phosphine ligands at 40°C , the formation of $\text{Ag}_2\text{Au}_{23}$ is possible. Since our work is focused on the synthesis of single-doped nanoparticles, this reaction is not presented in our manuscript.

-Since the nanoparticle has only two apex sites, it is not possible to achieve nanoparticles with more than 2 Ag. However, Cu can occupy both the waist and apex positions. Therefore, step-by-step doping of more than 2 Cu atoms is theoretically feasible. Nevertheless, our results show incorporation of more than 2 Cu atoms into the nanocluster decreases the stability of the doped particles that results in their decomposition.

There is ambiguity in the labelling of metal atom locations P3 and P4. In Figure 5 of the main paper, P3 is the waist positions while in Supplementary Figures 1 and 2, P3 is shown as the central position. This has to be clarified otherwise the whole discussion about the site preference becomes unclear.

Response: We thank the reviewer for catching this. It's our mistake! Supplementary Figures 1 and 2 are now revised accordingly:

Supplementary Figure 1. The crystal structure of $\text{Cu}_1\text{Au}_{24}$ nanorod, resolved in this work (left). Note, the Au_{25} nanocluster has five non-equivalent types of metal positions (P1 to P5). Our analysis shows sites P1 and P3 can be occupied by Cu or Au. Other sites (i.e., P2, P4, and P5) are found to be occupied *only* by Au. Of note, site P3 can be divided into 5 different groups according to the cluster's symmetry, which are shown in different colors (right). Only metal atoms are shown for clarity.

Supplementary Figure 2. The crystal structure of $\text{Ag}_1\text{Au}_{24}$ nanorod, resolved in this work (left). Note, the Au_{25} nanocluster has five non-equivalent types of metal positions (P1 to P5). Our analysis shows site P1 can be occupied by Ag or Au. Other sites (i.e., P2, P4, and P5) are found to be occupied *only* by Au. Of note, site P3 can be divided into 5 different groups according to the cluster's symmetry, which are shown in different colors (right). Only metal atoms are shown for clarity.

There is an error in the first paragraph of the section entitled, On the shuttling-in mechanism for the formation of MAu₂₄ (M=Cu or Ag). Figure 2 is wrongly referred in the sentence "The X ray crystallography analysis indicates that....(Fig 2)". In fact, Fig 2 does not contain any structures. It is not clear what the experimentally observed crystal structure in this work is. Are the structures shown in Figure 3 refer to the crystal structures found in this work?

Response: We thank the reviewer again. The mistake has been corrected! It should be Figure 3 and we have revised the manuscript accordingly. The Ag₁Au₂₄ and Cu₁Au₂₄ which are shown in Figure 3 do refer to the crystal structures found in this work.

In the captions of figures showing the structures (except that of Fig 1), it is not clearly mentioned whether they are crystal structures, resolved in this work, or a schematic made from reported coordinates. This has to be explicitly mentioned in the captions in main paper and in the SI.

Response: We have clarified that in the figure captions. Fig 2: spectra only, no structure; Fig 3, crystal structures from this work (except the hollow Au₂₄ from ref 35); Figs 4 and 5, calculated intermediates' structures.

In Figure 4, it is not explicitly mentioned whether the mechanism shown is from the DFT calculations performed in this work or not. In Figure 5, this is mentioned explicitly.

Response: In Fig 4, the mechanism is from DFT as in Fig 5. We have clarified it in the figure caption of Fig 4

It is not clear whether Supplementary Figures 1 and 2 show the experimentally resolved structures of Cu₁Au₂₄ and Ag₁Au₂₄ or just a general schematic showing the distinct metal atom locations.

Response: Both are crystal structure (Cu₁Au₂₄ and Ag₁Au₂₄) resolved in this work.

In Supplementary Figures 1 and 2, it is better to mention the positions of Cu and Ag. Also it has to be mentioned in the captions that (i) only metal atoms are shown, (ii) P1, P2, etc., are various symmetry unique metal atom positions. Also, the reader has no idea (i) what are the structures shown in the right of these figures conveying and (ii) why two structures are shown.

Response: We have changed the Figures/captions accordingly.

In Supplementary Figure 4, details of the energy differences to be explained in the caption. Also the color codes to be mentioned more clearly.

Response: We have done so.

In Supplementary Figure 5, color codes of atoms have to be mentioned explicitly. Also, it is advisable to mention that the figure shows a lateral/top view or something like that. It is mentioned in the caption that S atoms are green, however only one green sphere is shown, though there are more than five thiolate ligands.

Response: We have clarified these.

Reviewer #3 (Remarks to the Author):

In this manuscript, the authors have investigated atomically precisely doped metal nanoclusters and found that the Ag dopant goes to the apex site of the Au₂₄ rod while the Cu doping into both the apex and waist positions. Currently, the controlled preparation of doped nanomaterials is unusual. This paper provides valuable insights into the controlled doping both in position and number of heteroatoms in alloy nanoparticles, so will be of some interest to the on-going research of alloy nanoparticles. However, the methods employed in this study is similar to their previous article published in JACS 137, 4018 (2015), which may be qualified as a follow-up of earlier results on the single-atom doping into gold nanoparticles and lack of novelty to the readers. This difference of the Ag and Cu doping has been explained by the binding between the dopant and -Cl (-SR). I think the authors should also explain their difference with the Cd doping, for which the Cd is in the center of the 13-atom icosahedral core. Furthermore, although the alloying is often used to improve the properties of bulk materials, it is far more difficult to make this kind of doping in a nanostructured material due to the so-called "self-purification effect", leading to the expulsion of impurities. The authors should explain the reason why the alloy nanoclusters can be stable against the self-purification effect. The third question is that the transition states in Figs. 4 and 5 have been checked by the frequency calculations. In summary I think this work does not meet the criteria of novelty or urgency that apply to Nature Commun.

Response:

1) We thank the review for the comments. We believe the following two aspects distinguish our current work from the previous one:

(i) In the current study, we introduce a new method that is based on single vacancy formation prior to doping, i.e. hollowing first, then filling with a heterometal, which leads to the important results discussed in this manuscript. The previous paper in JACS involves a direct metal exchange method.

(ii) In the JACS paper, the Cd and Hg doping was exclusively of single-atom. However, for Ag and Cu doping, a **distribution** of Cu or Ag dopants was observed (see works of Negishi and other groups) and achieving single doping is a major challenge. Our new method has achieved it.

Per the reviewer's suggestion, we have tested the reaction of Au₂₄ with CdCl₂ salt (Figure R3, see below). Results show Cd cannot be incorporated into the nanocluster—The identical spectra indicate no replacement of Au to Cd atom. This is mainly due to the different electron configuration Cd (d¹⁰s²) in comparison to Au, Ag, and Cu (d¹⁰s¹). Our results in this work by devising the hollowing-refilling strategy, explain for the first time how to prepare gold nanoparticles doped with a hetero-metal from the same group of gold without the distribution.

Figure R3. UV-Vis spectra of solutions containing Au₂₄ nanocluster before (black line) and after addition of CdCl₂ for 30 mins (red line).

2) Regarding the reviewer's comment on "self-purification effect", such an effect was previously reported in larger nanoparticles. Those NPs were not perfect (e.g. with defects, unprotected surface sites, polycrystallinity, polydispersity, and so on) and their behavior would be complicated. In contrast, the well defined hollow Au₂₄ particles in the present work provide an ideal system for fundamental studies. The dopant-number-variable Au_{25-x}Ag_x (c.f., x up to 9 only in Negishi et al *Chem. Commun.*, 2010, 46, 4713 but x up to 19 in Li et al *Chem. Commun.*, 2016, 52, 5194) implied that the ligand might play some role in addition to those common factors invoked in the alloy NP research. The dopant # is much less in Au_{25-x}Cu_x (*Chem. Eur. J.* 2013, 19, 4238) which could be due to the "self-purification effect". Overall, there are many fundamentally interesting issues remained in bimetal NP research and we hope the atomically precise systems will offer insights into those issues in future research.

3) The frequency calculations are performed for the transition states, thanks for checking this!

Reviewers' comments:

Reviewer #2 (Remarks to the Author):

Regarding the work of Bakr et al., (Distinct metal-exchange pathways of doped Ag₂₅ nanoclusters *Nanoscale*, 2016, 8, 17333–17339), the authors state that, "An important finding of the work by Bakr et al. is that their method of metal exchange initially resulted in a distribution of AuxAg_{25-x} (x=1-8), in which the unstable AuxAg_{25-x} (x=2-8) nanoclusters gradually decomposed and only Au₁Ag₂₄ was detected in their crystallographic analysis.". I find that there is no such data available in their main paper and SI. In fact, the above mentioned *Nanoscale* paper deals with the conversion of [PdAg₂₄(SR)₁₈]₂- and [PtAg₂₄(SR)₁₈]₂- into [AuAg₂₄(SR)₁₈]- and [AuPtAg₂₃(SR)₁₈]₂-, respectively, and this work do not involve any AuxAg_{25-x} (x=1-8) clusters!!!! I think the authors are talking about another paper by Bakr et al. (Templated Atom-Precise Galvanic Synthesis and Structure Elucidation of a [Ag₂₄Au(SR)₁₈]₂ Nanocluster, *Angew. Chem. Int. Ed.* 2016, 55, 922 –926).

Of course atoms do behave as waves depending upon the type of experiments being performed. However, it is not appropriate to bring this aspect in the type of reactions mentioned here because it is easier to imagine reaction mechanisms in terms of breaking, making and rearrangements of bonds rather than a magical tunneling from outer sites to the inner sites of these type of clusters without affecting any bonds. In the absence of any experimental evidence or previous reports on such "tunneling" effects in metal cluster chemistry, the proposition of "tunneling" in metal atom substitution is really confusing the reader. Or, it is better the authors keep this suggestion with a little more detailed explanation (as written in their response) of the tunneling effect.

Spelling mistakes, "ocupioed" and "Othere", in the caption of Supplementary Figures 1 and 2.

In the Figure 3, it is better the authors label the middle structure also for the sake of completeness.

Still there is ambiguity in the labelling of metal atom locations in Supplementary Figures 1 and 2!!! In the captions of these Figures, the authors mention that the site 3 can be divided to 5 different groups (see the picture on the right side). Do the authors mean that the symmetry breaking of the P3 sites is due to the substitution of the Cu atom at the waist position? This must be clarified.

There are five symmetry non-equivalent sites (P1 to P5) for Au₂₅ nanorod. However, in Supplementary Figure 2, (crystal structure of Ag₁Au₂₄ nanorod resolved in this work), Ag atom occupy only the P1 site which reduces the symmetry of the Ag₁Au₂₄ molecule and there comes more than five unique sites.

Supplementary Figures 1 and 2 and their captions do not give a clear description of the crystal structures. The labelling of the metal atom sites in the structures is still confusing. I think the authors can present this data in an understandable fashion in the following way:

1. Give a general schematic of undoped Au₂₅ and Au₂₄ nanorod structures with symmetry nonequivalent metal sites clearly labeled. This can be a separate figure.
2. Give separate figures to show the crystal structures (one picture for one crystal structure) of Cu₁Au₂₄ and Ag₁Au₂₄. May be only the dopant atoms (Ag/Cu) can be given a different color and make all the 24 Au atoms with the same color. The authors may remove the pictures on the right side in the Supplementary Figures 1 and 2.

I see the authors have addressed all my previous concerns adequately, except a few minor points mentioned above. The manuscript can be accepted for publication, after addressing the above points, without any further revision.

Reviewer #3 (Remarks to the Author):

After modification, I think that this manuscript qualifies to be recommended for publication in Nature Communications.

Reviewer #2 (Remarks to the Author):

Regarding the work of Bakr et al., (Distinct metal-exchange pathways of doped Ag₂₅ nanoclusters *Nanoscale*, 2016, 8, 17333–17339), the authors state that, “An important finding of the work by Bakr et al. is that their method of metal exchange initially resulted in a distribution of Au_xAg_{25-x} (x=1-8), in which the unstable Au_xAg_{25-x} (x=2-8) nanoclusters gradually decomposed and only Au₁Ag₂₄ was detected in their crystallographic analysis.”. I find that there is no such data available in their main paper and SI. In fact, the above mentioned *Nanoscale* paper deals with the conversion of [PdAg₂₄(SR)₁₈]²⁻ and [PtAg₂₄(SR)₁₈]²⁻ into [AuAg₂₄(SR)₁₈]⁻ and [AuPtAg₂₃(SR)₁₈]²⁻, respectively, and this work do not involve any Au_xAg_{25-x} (x=1-8) clusters!!!! I think the authors are talking about another paper by Bakr et al. (Templated Atom-Precise Galvanic Synthesis and Structure Elucidation of a [Ag₂₄Au(SR)₁₈]⁺ Nanocluster, *Angew. Chem. Int. Ed.* 2016, 55, 922 –926).

Response: We have cited both works in the revised manuscript. For the *ACIE* paper, the doping results suggest the Au_xAg_{25-x} with x=2-8 nanoclusters were gradually decomposed and only Au₁Ag₂₄ was found in the final product. In their *Nanoscale* paper, using PdAg₂₄ instead of Ag₂₅ nanocluster as starting material, the final product was mainly Au₁Ag₂₄. Together with the Pt doping, all the results suggest the importance of central doping in the metal exchange process.

Revision: Another interesting finding by Bakr and co-workers is that the single Pd atom in the Pd₁Ag₂₄ nanocluster could be replaced by gold atom, resulting in the Au₁Ag₂₄ nanocluster.³⁰

New Ref 30. Bootharaju, M. S., Sinatra, L., Bakr, O. M. Distinct metal-exchange pathways of doped Ag₂₅ nanoclusters. *Nanoscale*, 8, 17333–17339 (2016).

Of course atoms do behave as waves depending upon the type of experiments being performed. However, it is not appropriate to bring this aspect in the type of reactions mentioned here because it is easier to imagine reaction mechanisms in terms of breaking, making and rearrangements of bonds rather than a magical tunneling from outer sites to the inner sites of these type of clusters without affecting any bonds. In the absence of any experimental evidence or previous reports on such “tunneling” effects in metal cluster chemistry, the proposition of “tunneling” in metal atom substitution is really confusing the reader. Or, it is better the authors keep this suggestion with a little more detailed explanation (as written in their response) of the tunneling effect.

Response: ‘Quantum’ and ‘magical’ are different. ‘Quantum tunneling’ has its firm basis in wavefunction (e.g. ψ extension into the solution at a finite probability, rather than ‘magical’!

Revision: In the Abstract and the paragraph above ‘Results’, we have deleted “via tunneling”.

Spelling mistakes, “ocupoied” and “Other”, in the caption of Supplementary Figures 1 and 2.

Response: We have corrected them.

In the Figure 3, it is better the authors label the middle structure also for the sake of completeness.

Response: We have redrawn this figure.

Figure 3. Shuttling one Ag or Cu atom into the 24-atom hollow gold nanoparticle: pathways of single Ag/Cu entering the hollow Au₂₄ nanoparticle. Note, Ag₁Au₂₄ and Cu₁Au₂₄ are presented using X-ray crystallographic data of

this work, and Au₂₄ is shown according to the structure given in ref. 33. Color codes: Au, yellow; Ag, blue; Cu, magenta. Other non-metal atoms and bonds are not shown for clarity.

Still there is ambiguity in the labelling of metal atom locations in Supplementary Figures 1 and 2!!! In the captions of these Figures, the authors mention that the site 3 can be divided to 5 different groups (see the picture on the right side). Do the authors mean that the symmetry breaking of the P3 sites is due to the substitution of the Cu atom at the waist position? This must be clarified.

Response: The symmetry breaking of the P3 sites is not due to the substitution of the Cu atoms. In undoped Au₂₅ nanocluster, the gold atoms at P3 site are not equal (new Supplementary Figure 1D). We have redrawn the Supplementary Figures 1 and 2.

New Supplementary Figure 1. The crystal structure of undoped A) Au₂₄ and B) Au₂₅ nanorod; C) the metal core of Au₂₅ nanorod, the layer by layer gold atoms are labeled from P1 to P5; D) the P3 positions are divided into 5 different groups according to the cluster's symmetry.

There are five symmetry non-equivalent sites (P1 to P5) for Au₂₅ nanorod. However, in Supplementary Figure 2, (crystal structure of Ag₁Au₂₄ nanorod resolved in this work), Ag atom occupy only the P1 site which reduces the symmetry of the Ag₁Au₂₄ molecule and there comes more than five unique sites.

Supplementary Figures 1 and 2 and their captions do not give a clear description of the crystal structures. The labelling of the metal atom sites in the structures is still confusing. I think the authors can present this data in an understandable fashion in the following way:

1. Give a general schematic of undoped Au₂₅ and Au₂₄ nanorod structures with symmetry nonequivalent metal sites clearly labeled. This can be a separate figure.
2. Give separate figures to show the crystal structures (one picture for one crystal structure) of Cu₁Au₂₄ and Ag₁Au₂₄. May be only the dopant atoms (Ag/Cu) can be given a different color and make all the 24 Au atoms with the same color. The authors may remove the pictures on the right side in the Supplementary Figures 1 and 2.

Response: We have adjusted the Supplementary figures.

New Supplementary Figure 2. The crystal structure of $\text{Cu}_1\text{Au}_{24}$, resolved in this work. Our analysis on $\text{Cu}_1\text{Au}_{24}$ nanorod shows that sites P1 and P3 can be occupied by Cu. Other sites (i.e., P2, P4, and P5) are found to be occupied only by Au. Of note, site P3 can be divided into 5 different groups according to the cluster's symmetry. Only metal atoms are shown for clarity.

New Supplementary Figure 3. The crystal structure of $\text{Ag}_1\text{Au}_{24}$, resolved in this work. For $\text{Ag}_1\text{Au}_{24}$ Our analysis shows that site P1 can be occupied by Ag. Other sites (i.e., P2, P3, P4, and P5) are found to be occupied *only* by Au. Only metal atoms are shown for clarity.